



1. Title
pH-based ecological coherence of active canonical methanotrophs in paddy soils
2. Author Names
Jun Zhao, Yuanfeng Cai, Zhongjun Jia[*]
3. Author Affiliations
State Key Laboratory of Soil and Sustainable Agriculture
Institute of Soil Science, Chinese Academy of Sciences
Nanjing, 210008, Jiangsu Province, China
4. Corresponding author[*]
Zhongjun Jia; E-mail: jia@issas.ac.cn; Tel: +86-25-8688-1311



**Abstract**

Soil pH is considered one of the main determinants of the assembly of globally
distributed microorganisms that catalyse the biogeochemical cycles of carbon and
nitrogen. However, direct evidence for niche specialization of microorganisms in
association with soil pH is still lacking. Using methane-oxidizing bacteria
(methanotrophs) as a model system of carbon cycling, we show that pH is potentially
the key driving force selecting for canonical gamma- (type I) and alpha- (type II)
methanotrophs in rice paddy soils. DNA-based stable isotope probing (DNA-SIP) was
combined with high-throughput sequencing to identify the taxonomic identities of
active methanotrophs in physiochemically contrasting soils from 6 different paddy
fields across China. Following microcosm amendment with $^{13}CH_4$, methane was
primarily consumed by *Methylocystis*-affiliated type II methanotrophs in soils with a
relatively low pH (5.44-6.10), whereas *Methylobacter*/*Methylosarcina*-affiliated type I
methanotrophs dominated methane consumption in soils with a high pH (7.02-8.02).
Consumption of $^{13}CH_4$ contributed 0.203% to 1.25% of soil organic carbon, but no
significant difference was observed between high-pH and low-pH soils. The
fertilization of ammonium nitrate resulted in no significant changes in the compositions
of $^{13}C$-labelled methanotrophs in the soils, although significant inhibition of methane
oxidation activity was consistently observed in low-pH soils. Mantel analysis further
validated soil pH, rather than other parameters tested, had significant correlation to the
variation of active methanotrophic compositions across different rice paddy soils. These
results suggest that soil pH might have played pivotal roles in mediating the niche
differentiation of ecologically important aerobic methanotrophs in terrestrial
ecosystems and imply the importance of such niche specialization in regulating
methane emissions in paddy field under increasingly intensified input of anthropogenic
N fertilizers.
**Keywords:** pH; Niche differentiation; Methanotrophs; Rice paddy; DNA stable isotope
probing; High-throughput sequencing



## 1 Introduction

Rice paddy fields are one of the major sources of the potent greenhouse gas methane, contributing to approximately 10-25% of global methane emission (Kögel-Knabner et al., 2010). Constantly produced through methanogenesis from the anaerobic compartment of inundated paddy fields, methane can diffuse into the oxic-anoxic interface of the soil and reach high concentrations of above 5 mM, which is approximately equivalent to 50,000 ppmv (Eller and Frenzel, 2001;Nouchi et al., 1990;Nouchi et al., 1994). It is estimated that up to 80% of methane gas is consumed by soil aerobic methanotrophs (i.e. methane-oxidizing bacteria, MOB) before being released into the atmosphere (Conrad and Rothfuss, 1991;Frenzel et al., 1992). Therefore, methanotrophs in rice paddy fields are considered a crucial biological filter attenuating potential methane emissions as well as an important contributor to the maintenance of soil organic carbon (C) because a substantial amount of $CH_4$-derived C was used for MOB grow and biomass synthesis (Bridgham et al., 2013).

Accumulating evidence has indicated that the specialized MOB clades designated as upland soil clusters α and γ (USCα and USCγ, respectively) have high $CH_4$ affinity and catalyse atmospheric methane oxidation in unsaturated soils (Holmes et al., 1999;Roslev and Iversen, 1999;Tveit et al., 2019;Knief et al., 2003), while the canonical *α-* and *γ-Proteobacterial* MOB (known as type II and type I methanotrophs) are considered to adapt better to high concentration methane and thus regulate the methane oxidation in ecosystems with constant methane productions, such as wetlands. Despite co-existence of type I and II methanotrophs in these ecosystems, their activity and contribution to methane oxidation vary largely depending on different environmental conditions and the predominant activity of either type I or type II methanotrophs has often been observed (Daebeler et al., 2014;Liebner and Wagner, 2007;He et al., 2012;Chen et al., 2008a;Lin et al., 2004). This might be due to some major physiological differences that exist between these two groups. For instance, type I methanotrophic strain is more competitive under relatively lower methane and higher oxygen concentrations compared to type II methanotrophs (Graham et al., 1993).



Additionally, adaptation to slightly acidic pH values (growth optima 5.0 - 6.0) is
characteristic for type IIb (*Methylocella* and *Methylocapsa*) and some *Methylocystis*
strains (Dedysh et al., 2000;Dedysh et al., 2007;Belova et al., 2013). These and other
physiological traits of type I and II methanotrophs may be important in partitioning
their specialized niches in different ecosystems.
Increasing lines of evidence from ecological studies have suggested that soil pH,
among other environmental variables, might be one of the most important determinant
in the emergence and maintenance of microbial communities across a wide variety of
environments (Tripathi et al., 2018;Lauber et al., 2009). Recent studies have provided
compelling evidence for niche specialization of biogeochemically important guilds
associated with pH variation and consequent distinct patterns of soil resource utilization.
For example, the biogeographical distribution of ammonia-oxidizing oxidizers is more
strongly associated with soil pH than other parameters tested in soils (Gubry-Rangin et
al., 2011;Aigle et al., 2019), as is that of denitrifiers (Liu et al., 2010). It is implied that
type I and II methanotrophs might also be selectively favoured under different pH
conditions in natural wetland system, despite no systematic comparison has yet been
made. Dominance of type II methanotrophs have been revealed in many natural acidic
peatlands (Chen et al., 2008a;Dedysh, 2009;Kip et al., 2012;Chen et al., 2008b;Gupta
et al., 2012), whereas in neutral-alkaline wetlands, type I methanotrophs appear to be
more active (Lin et al., 2004;Gupta et al., 2012;Morris et al., 2002). The similar pattern,
however, has not been obtained in anthropogenically flooded rice paddy fields, possibly
due to lack of study on methanotrophic activity in such soils with a wide range of pH.
In fact, type I methanotrophs were considered to dominantly catalyse methane at high
concentration in the rice paddy soils (Ma et al., 2013;Qiu et al., 2008;Reim et al.,
2012;Shrestha et al., 2008), while the activity of type II methanotrophs and their
contribution to methane oxidation in rice soils remains unclear (Semrau et al.,
2011;Leng et al., 2015). Since these findings all came from neutral-alkaline soils, it is
necessary to investigate the active methanotrophs in more acidic soils, which likely
have different community compositions (Shiau et al., 2018).



In this study, we selected 6 rice paddy soils with a pH gradient ranging from 5.45 to 8.02 collected from 6 geographically different rice fields located in the main rice production areas across the south, east and middle of China, including Yu-Xi (YX), Ying-Tan (YT), Tao-Yuan (TY), Zi-Yang (ZY), Chang-Shu (CS) and Lei-Zhou (LZ) (Fig. S1) and used DNA-based stable isotope probing (DNA-SIP) to identify active methanotrophs under unfertilized situation and following a simulated fertilization. We predict distinct compositions of methanotrophic phylotypes of type I and type II in these soils which are associated with soil pH. Other environmental factors pertinent to methane oxidation were also tested to further elaborate the importance of soil pH in selection of active methanotrophic phylotypes in the rice paddies.

## 2 Material and methods

### 2.1. Site description and soil sampling

The soils were collected from 6 different rice fields located in the main rice production areas across the south, east and middle of China, including Yu-Xi (YX), Ying-Tan (YT), Tao-Yuan (TY), Zi-Yang (ZY), Chang-Shu (CS) and Lei-Zhou (LZ) (Fig. S1). All sites have a subtropical or tropical monsoon climate and a history of rice cultivation for >50 years. The fields usually receive annual fertilization of 250 to 350 kg N ha$^{-1}$. Soil sampling was performed at a 0-20-cm depth by mixing at least three randomly collected soil cores. The composite soil samples were air-dried as previously described (Noll et al., 2008;Mohanty et al., 2006;Murase and Frenzel, 2007a) and passed through a 2.0-mm-pore-size sieve before the construction of microcosms.

### 2.2. Soil physiochemical properties

The pH was assessed by a Mettler Toledo 320-S pH meter (Mettler Toledo Instruments, China) with a water-to-soil ratio of 2.5. Soil inorganic N (ammonium plus nitrate) was extracted from soil with a 2 M KCl solution and quantified using a Skalar San Plus segmented flow analyser (Skalar, The Netherlands). The soil organic matter (SOM) content was determined using the dichromate oxidation method. Total organic



carbon (TOC) and total N (TN) were determined by a vario Max CN Element Analyzer
(Elementar, Germany). The available soil copper content was determined using an
OPTIMA 8000 inductively coupled plasma optical emission spectroscope (ICP-OES)
(PerkinElmer,    USA)    after    extraction    with    buffered    5    mM
Diethylenetriaminepentaacetic acid (DTPA) solution. The soil oxidation capacity (OXC)
was determined using the equation $5 \times [NO_3^-] + 2 \times [Mn(IV)] + [Fe(III)] + 8 \times [SO_4^{2-}]$ as
detailed previously (Zhang et al., 2009). The results of these properties were shown in
Table 1.
**2.3. Stable-isotope probing of methane-oxidizing bacteria**

Three treatments were established in triplicate, including "Control" (under natural

atmospheric condition), "$^{13}CH_4$" (incubated with 5% v/v $^{13}CH_4$ supplementation) and
"$^{13}CH_4$+N" (incubated with 5% v/v $^{13}CH_4$ supplementation plus fertilization with
$NH_4NO_3$). Soil equivalent to 6.0 g *d.w.s.* was incubated at a maximum water-holding
capacity of approximately 60% and at 28°C in the dark in a 120-ml serum bottle sealed
with a butyl stopper. For the "$^{13}CH_4$" and "$^{13}CH_4$+N" treatments, 6 ml of the headspace
air in the bottles was replaced with the same volume of >99% pure $^{13}CH_4$ gas
(Cambridge Isotope Laboratories, USA) to make an initial methane mixing ratio of
approximately 5% in the headspace. For the "$^{13}CH_4$+N" treatment, $NH_4NO_3$ solution
instead of distilled water was added to the soil microcosm in a dropwise manner to
produce a supplement of 200 µg inorganic N g soil$^{-1}$. A 2-day pre-incubation was
performed before applying $^{13}C$-labelled methane and nitrogen fertilizer.

For the "$^{13}CH_4$" and "$^{13}CH_4$+N" treatments, the headspace methane mixing ratios

were measured every two days by an Agilent 7890A Gas Chromatograph (Agilent
Technologies, USA) to assess the rate of methane oxidation in SIP microcosms. Each
microcosm incubation was completed when approximately 90% of the methane gas was
consumed, i.e., the headspace methane concentration dropped to below 5,000 ppmv, or
after 6 weeks if the headspace methane concentration was still higher than 5,000 ppmv.
Soils were then collected and stored at -80°C for further analyses.



### 2.4. Soil $^{13}$C-atom abundance assay

The synthesis of biomass carbon derived from $^{13}$C-CH$_4$ was assessed by determination of $^{13}$C-atom abundance in soil organic matter. Approximately 1.5 g of each frozen soil sample was vacuum freeze-dried using an Alpha 1-2 LDplus freeze dryer (Christ, Germany). The relative $^{13}$C-atom ratio was assessed by a Flash 2000 elemental analyser coupled to a Delta V Advantage isotope ratio mass spectrometer (Thermo Scientific, USA), and the TOC content was then measured by a vario Max CN Element Analyzer (Elementar) using the desiccated soil samples.

### 2.5. DNA extraction and SIP gradient fractionation

Total DNA was extracted using 0.5 g of each soil by a FastDNA spin kit for soil (MP Biomedicals, USA) according to the manufacturer's instructions. The quantity and quality of DNA extracts were assessed using a NanoDrop ND-1000 UV-visible light spectrophotometer (NanoDrop Technologies, USA).

The isopycnic density gradient centrifugation was employed to $^{13}$C-DNA from $^{12}$C-DNA in the total DNA extract. In brief, approximately 2.5 µg of the extracted DNA was mixed with a CsCl solution to achieve a final volume of 5.5 ml with a CsCl buoyant density of 1.725 g ml$^{-1}$ following ultracentrifugation at 177,000 g and 20°C for 44 h in a Vti65.2 vertical rotor (Beckman Coulter, USA). The DNA fractions for each sample were collected and measured for CsCl density as previously described (Zhao et al., 2015;Wang et al., 2015). The fractionated DNA was purified with 70% ethanol after polyethylene glycol (PEG) 6000 precipitation and dissolved in 30 µl of sterile water.

### 2.6. Real-time quantitative PCR of biomarker *pmoA* genes

To determine the changes in abundance of methanotrophic communities and to assess the $^{13}$CH$_4$ labelling of methanotrophs, the copy number of *pmoA* genes in the total DNA extracts as well as in the DNA gradient fractions (fractions 3-13) were determined by real-time quantitative PCR (qPCR) using a CFX96 Optical Real-Time detection system (Bio-Rad Laboratories, USA). The PCR primers A189f/mb661r were used (Holmes et al., 1995;Costello and Lidstrom, 1999) following the conditions shown





in Table S1. The standards were generated using plasmid DNA from one representative
clone containing bacterial *pmoA* genes, and a dilution series of standard template from
$10^2$ to $10^8$ per assay was used. In addition, the total DNA extracts were diluted in a
series to assess possible PCR inhibition by soil humic substances, and DNA extracts
were diluted 20-fold for subsequent analysis. The amplification efficiencies ranged
from 92% to 103%, with $R^2$ values of 0.994 to 0.999. Melting curve analysis and
standard agarose gel electrophoresis were always performed at the end of a PCR run to
verify the amplification specificity.

**2.7. MiSeq sequencing of 16S rRNA and *pmoA* genes**

Illumina MiSeq sequencing was employed to investigate the community shifts of
methanotrophs in the soils. The total microbial communities were analysed in all soil
microcosms using universal primers for 16S rRNA genes to investigate the proportional
changes in methanotrophs relative to the total microbial communities in soils. In
addition, the $^{13}$C-DNA retrieved from "heavy" CsCl fractions (with a density of 1.738-
1.740 g ml$^{-1}$) in $^{13}$CH$_4$-labelled microcosms was also subjected to amplicon-based
sequencing targeting both the 16S rRNA and *pmoA* genes. The "light" DNA fractions
(with a density of 1.719-1.726 g ml$^{-1}$) from the $^{12}$C-control samples were also used for
16S rRNA gene sequencing to reveal the background microbial community
compositions. The PCR primer pairs were 515F/907R (Stubner, 2002) for 16S rRNA
genes and A189f/mb661r for *pmoA* genes, with each forward primer fused with a
unique barcode sequence. The PCR primers and conditions are detailed in Table S1.
The resulting PCR products were gel purified and combined in equimolar ratios in a
single tube. The sequencing samples were prepared using a TruSeq DNA kit (Illumina,
USA), and the purified library was diluted, denatured, re-diluted, and mixed with PhiX
as described in the Illumina library preparation protocols. Paired-end sequencing
(2×300 bp) was conducted using the Illumina MiSeq system (Illumina, USA).

**2.8. Sequence data processing and deposition**



All raw sequence files were processed using the Quantitative Insights Into
Microbial Ecology (QIIME) pipeline (Caporaso et al., 2010). Paired-end reads were
first assembled using FLASH with a minimum overlap parameter value of 10 bp
(Magoc and Salzberg, 2011). The quality control procedure removed reads with a
quality score <20, mismatched primers and ambiguous bases. Chimeras were
eliminated using USEARCH. For *pmoA* genes, putative frame-shifting reads were
removed using the FRAMEBOT program (Wang et al., 2013). Subsequently, a total of
2,871,893 high-quality 16S rRNA and 421,696 *pmoA* sequences were retained for
further analyses. The high-quality sequences were then clustered into operational
taxonomic units (OTUs) at 97% (16S rRNA gene) or 93% (*pmoA* gene) sequence
similarities by the UPARSE algorithm (Edgar, 2013). To cluster *pmoA* genes at 93%
similarity, the "otu_radius_pct" option (default of 3) was changed to 7 when performing
the "cluster_otus" command, and the id option (default of 0.97) was modified to 0.93
for the "usearch_global" command. The representative sequences of all 16S rRNA
OTUs were taxonomically classified using the Ribosomal Database Project (RDP)
classifier (Wang et al., 2007). For *pmoA* reads, representative OTU sequences were
classified using a naïve classifier implemented with the mothur "classify.seq" command
as described previously (Dumont et al., 2014). For the major $^{13}$C-labelled *pmoA* OTUs
retrieved from heavy CsCl fractions (containing ≥2% of *pmoA* gene sequences in at
least one of the samples), a representative sequence was selected for phylogenetic
analysis by comparison with known sequences from GenBank. A heatmap was
constructed based on the relative abundances of the major $^{13}$C-labelled *pmoA* OTUs
across different microcosms by HemI version 1.0 (Deng et al., 2014), and hierarchical
clustering of samples was performed with the calculated Pearson distance.
**2.9. Statistical analysis**
Mantel tests were used to test for significant correlations between methanotrophic
community distance and different soil physiochemical properties, namely, pH, TOC,
TN, C:N ratio, SOM, exchangeable inorganic N, soil copper content and OXC. The
tests were performed in the *R* environment with the vegan package (Dixon, 2003).




One-way analysis of variance with Tukey's *post hoc* test was used for comparisons
among different treatments for each soil. An independent t-test was conducted to assess
the possibility of significant differences between two groups. Analyses were conducted
using the SPSS version 13.0 package for Windows (SPSS, Inc.). $P<0.05$ was regarded
as statistically significant.

## 3   Results

### 3.1. Conversion of $CH_4$ to soil organic matter

During SIP microcosms incubation a fraction of $^{13}C$-$CH_4$ were converted to soil
organic matter (SOM) by MOB through cell biomass synthesis, and it was assessed as
the changes in the $^{13}C$-atom percent of the soil total organic carbon (TOC). The average
background $^{13}C$-atom abundance in soils under natural atmospheric condition was
$1.08\pm0.01\%$, and all soils showed statistically significant increases in the $^{13}C$-atom
abundance of TOC up to $1.74\pm0.41\%$ upon consumption of $^{13}CH_4$ (Fig. 1a). There was
no significant difference in conversion ratio of $^{13}CH_4$ into soil organic matters between
low-pH and high-pH soils, and fertilizing soils with inorganic nitrogen did not result in
higher $^{13}C$ incorporation into organic matter than that in unfertilized soils (Table S2).
Based on the changes in methane concentrations, soil organic carbon contents and $^{13}C$-
atom percent during incubation of SIP microcosms, it was theoretically estimated that
10.4-38.1% of $^{13}C$-$CH_4$ was converted to soil organic carbon during the microcosm
incubations (Table S2). And $^{13}CH_4$-derived carbon contributed 0.203-1.25% of total soil
organic carbon after incubation (Table S2).

### 3.2. Methane oxidation rates

Assuming linear kinetics, the methane oxidation rates were 0.71-4.08 μmol $CH_4$ g
$d.w.s.^{-1}$ day$^{-1}$ in soils of YX, YT and TY (with a low pH) and 2.65-4.83 μmol $CH_4$ g
$d.w.s.^{-1}$ day$^{-1}$ in soils of ZY, CS and LZ (with a high pH) (Fig. 1b). Nitrogen fertilization
led to significantly lower methane oxidation rates in low-pH soils, while varying effects
were observed in high-pH soils (Fig. 1b). The changes in the concentrations of



headspace methane in the microcosms also showed inhibition by inorganic nitrogen of
microbial methane oxidation during incubation of low-pH soils, leading to a prolonged
period for consumption of the same amount of methane, particularly for YT soil (Fig.
S2).
**3.3. Population dynamics of methane-oxidizing bacteria**
The absolute abundance of soil methanotrophs was estimated by qPCR using the
biomarker gene *pmoA* at the microcosm incubation endpoints. The consumption of
methane at high concentrations stimulated the growth of methanotrophs, represented
by 20.4- to 1,027-fold increases in the number of *pmoA* gene copies following
microcosm incubation in all six soils (Fig. 1c). Consistent with the dynamic changes in
methane oxidation rates, nitrogen fertilization had a similar impact on the abundance
of methanotrophic communities. Particularly, the *pmoA* gene abundances under
nitrogen fertilization were significantly lower than those in the unfertilized low-pH
soils (YX, YT and TY) (Fig. 1c).
Based on high-throughput sequencing of 16S rRNA genes, the percentages of type
I and type II MOB in the total microbial communities were calculated to track the
dynamic changes of methanotrophic communities during SIP incubations. The relative
abundances of type II methanotrophs increased significantly in methane-amended
microcosms with low-pH soils (YX, YT and TY) but did not change in the high-pH
soils of ZY, CS and LZ (Fig. 1d). For type I methanotrophic populations, the reversed
trend was observed. Specifically, type Ia methanotrophs showed an 8.0-16.9-fold
increase in high-pH soils of ZY, CS and LZ, while no change or only a minor increase
was observed in the low-pH soils (YX, YT and TY) following methane amendment
(Fig. 1e). High enrichment of type Ib methanotrophs was also observed in the ZY soil
(Fig. 1f).
**3.4. Stable isotope probing of active methane-oxidizing bacteria**
Following the isopycnic centrifugation of the total DNA extracted from $^{13}CH_4$-SIP
microcosms, real-time qPCR analysis of *pmoA* genes as a function of the buoyant





density demonstrated active cell propagation and $^{13}$C assimilation in all six soils fuelled
by methane oxidation. A peak shift of relative abundances of *pmoA* genes towards
heavy fractions was clearly observed in all soil microcosms amended with $^{13}$CH$_4$
compared to the control treatments (Fig. 2). The *pmoA* genes in the $^{13}$CH$_4$-amended
microcosms were highly accumulated in the heavy DNA fractions with a buoyant
density of approximately 1.735-1.745 g ml$^{-1}$, while the *pmoA* genes in the control
treatments peaked only in the light DNA fractions with a buoyant density of 1.717-
1.726 g ml$^{-1}$. Similar results were also obtained for SIP microcosms amended with
inorganic nitrogen (Fig. 2). Notably, following N fertilization, the highest peak in the
YT soil occurred in the light fraction, although the apparent labelling of *pmoA* gene-
carrying methanotrophs was evidenced by increased abundances in the heavy fraction
compared to control.

High-throughput sequencing of the 16S rRNA genes in the heavy DNA fractions

at the whole-community level further showed significant increase in relative abundance
of methanotrophs in $^{13}$C-labelled microcosms. The methanotroph-affiliated $^{13}$C-16S
rRNA genes accounted for 61.9 to 81.2% of the total microbial communities in the $^{13}$C-
DNA fractions, while in the control treatment, the background methanotrophs
constituted only 3.1-7.2% of the total microbial communities (bottom columns of Fig.

2).

**3.5. Linking soil physiochemical properties with active methanotrophs**

Phylogenetic analysis of $^{13}$C-labelled *pmoA* genes from heavy DNA fractions

demonstrated that *Methylocystis* related type II organisms dominated the $^{13}$C-labelled
methanotrophs in the YX, YT and TY soils with low pH values (Fig. 3), which was
confirmed by taxonomic classification of $^{13}$C-labelled 16S rRNA gene classification
(Fig. S3 and S4). In the high-pH soils (ZY, CS and LZ), $^{13}$C-labelled methanotrophs
were predominated by type Ia organisms. The $^{13}$C-*pmoA* genes were related to the type
Ia methanotroph *Methylobacter* sp. (Fig. 3), but the $^{13}$C-labelled 16S rRNA genes
suggested the methanotrophs might be closer to *Methylosarcina* sp. (Fig. S4). Notably,
in the ZY soil, 17-30% of the $^{13}$C-labelled sequences were phylogenetically related to



the type Ib methanotroph *Methylocaldum* sp. (Fig. 3). The community compositions of
the $^{13}$C-labelled methanotrophs were deeply branched between high- and low-pH soils
(Fig. 3).

Mantel tests showed that only the pH, out of all the eight soil characteristics tested,

was significantly correlated with variation in the active methanotrophic communities
(according to $^{13}$C-labelled *pmoA* gene sequencing) between different soils ($P<0.05$)
(Table S3). Regression analysis further revealed a significantly positive relationship
between soil pH and the relative abundances of the primary $^{13}$C-labelled type I
methanotrophic cluster (OTU17), while pH was negatively related to the relative
abundances of the dominant type II cluster (OTU32) under high methane concentrations
(Fig. 4).

## 4   Discussion

Our results provide strong evidence for the important roles of pH-based selection

of type I and type II methanotrophs in methane oxidation and assimilation occurring in
paddy soils. In soils amended with $^{13}CH_4$ gas, the incorporation of $^{13}$C into genomic
DNA occurred in methanotrophic communities that directly utilized methane-derived
carbon for growth. Therefore, the relative gene abundances of type I versus type II
methanotrophs in $^{13}$C-labelled DNA fractions can reflect their relative contributions to
actual methane uptake and oxidation. In this study, the ratios of $^{13}$C-labelled type II to
type I methanotrophs in ZY, CS and LZ soils with high pH values were very low ranging
from 0.002 to 0.014 (Table S4), suggesting that the type I methanotrophs in these three
soils were more active than their type II counterparts. However, the ratios of $^{13}$C-
labelled type II to type I were 20.0 to 101 in the in the low-pH soils of YX, YT and TY,
respectively (Table S4). Assuming that one cell contained 2 copies of *pmoA* genes (Kolb
et al., 2003), type II methanotrophs could reach a cell-specific methane oxidation rate
of 0.2-9.6 fmol $CH_4$ h$^{-1}$ cell$^{-1}$ in these three soils (calculated in Table S4), being
consistent with previous reports 0.2-15 fmol $CH_4$ h$^{-1}$ cell$^{-1}$ obtained from both pure
cultures and complex soils (Hanson and Hanson, 1996). These results thus suggest that



aerobic methanotrophy was mostly sustained by the growth of type II methanotrophs
in these acidic paddy soils.
Type II methanotrophs identified in low-pH soils were affiliated to *Methylocystis*
based on [13]C-labelled *pmoA* genes (Fig. 3), which was also congruent with phylogenetic
analysis of 16S rRNA genes (Fig. S4). The analysis of *pmoA* genes indicated that [13]C-
methanotrophs in high-pH soils could be most closely related to *Methylobacter* of type
Ia methanotrophs. However, phylogenetic analysis of the [13]C-labelled 16S rRNA genes
suggested that in high-pH soils the dominant methanotrophs could cluster closely with
*Methylosarcina* species (Fig. S4). We assume this discrepancy might be attributed to
the lack of whole genome information in the present study, and the presence of
phylogenetic incongruence between *pmoA* and 16S rRNA genes which could be better
resolved with the increasing genome availability of methanotrophs in the future.
Nevertheless, our results (both from *pmoA* and 16S rRNA genes) indicated that type Ia
methanotrophs dominated methane oxidation activities in the high-pH paddy soils.
It is technically challenging for tracking the *in situ* activities of microbes, especially
in ecosystems that are exposed to constantly fluctuating environments such as rice
paddy fields. Agricultural management might influence the population dynamics of
methanotrophs through irrigation, fertilization and plantation. In view of this variability,
results based on microcosms cannot represent entirely the *in situ* conditions. However,
the incubation of SIP microcosms was conducted under the same conditions, but the
labelling of distinct methanotrophs indeed occurred, which might be the result of long-
term ecological and evolutionary adaption of active methane oxidizers in paddy soils
with contrasting physiochemical variables. Therefore, our results might reflect what is
largely occurring under *in situ* inundated conditions, in which high methane emissions
occur, particularly in regard to the relative activities of type I versus type II
methanotrophs that were likely controlled by soil intrinsic biotic and abiotic factors.
The active methanotrophic compositions in the six paddy soils were strongly
associated with only soil pH, based on tests on potential correlations with 8 key soil
physiochemical properties. Specifically, rice paddy soils with higher type II





methanotrophic activities were all acido-neutral (YX, YT and TY, with a pH of 5.44-
6.10), while the more alkaline soils (ZY, CS and LZ, with a pH of 7.02-8.02) displayed
stronger type I methanotrophic activity. Previous culture-dependent and
ecophysiological studies, where type II methanotrophs were described as stress
tolerators, provided strong support for low pH as the potential driving force for
selection of type II over type I methanotrophs in paddy soils (Ho et al., 2013).
Cultivated acidophilic or acid-tolerant methanotrophs are by far mostly type II strains
which are phylogenetically close to the $^{13}$C-methanotrophs retrieved in the low-pH soils
in this study (Fig. 3). Our results are consistent with the observations from natural
wetland systems, in which the activity of type II methanotrophs appeared to be more
prevalent under low-pH than high-pH conditions (Chen et al., 2008a;Dedysh, 2009;Kip
et al., 2012;Chen et al., 2008b;Gupta et al., 2012). It thus seems plausible that oxidation
of methane at high concentrations by type II methanotrophs more frequently occurs in
acidic than alkaline rice soils. However, the possibility cannot be ruled out that other
untested abiotic and biotic variables might have stronger forces in shaping community
structure of methanotrophs in rice paddy soils.
Our results demonstrated that chemical nitrogen fertilization did not alter the
dominant community compositions of active methanotrophs (Fig. 3), implying that the
growth of methanotrophs in these soils was not nitrogen-limited, or nitrogen availability
was not the key factor for selection of distinct methanotrophs in rice soils. In this
context, we speculate that N input could have stimulated plant growth and increased
the levels of exudates and litter decomposition served as precursors for methanogenesis
to enhance methane production. Despite no consistent pattern was observed with
respect to the effects of N fertilization on methane oxidation rates in this study, it
appears that N fertilization had an inhibitory effect on the methane oxidation rates in
the low-pH soils, which were dominated by type II methanotrophic activity (Fig. 1b
and Fig. S2). Consistently, the methanotrophic abundances under N fertilization
situation were significantly lower than those in unfertilized low pH soils (Fig. 1c).
These results agreed well with previous findings that high nitrogen input suppressed



the activity and growth rates of type II methanotrophs from pure cultures (Graham et
al., 1993) and in complex soil samples (Noll et al., 2008;Bodelier et al., 2000;Mohanty
et al., 2006). Meanwhile it is noteworthy that contradictory results were obtained in
high-pH paddy soils. N fertilization stimulated the methane oxidation rate in only one
of the high-pH soils (LZ), which had a unique dominant type Ia cluster (OTU50, as
shown in Fig. 3) when compared to the other two high-pH soils (dominated by OTU17).
Our study therefore implied that the contradicted effects of nitrogen fertilization on
methane oxidation (inhibition or stimulation) frequently reported in different soils
(Bodelier et al., 2000;Zheng et al., 2014;Alam and Jia, 2012;Cai and Yan, 1999) might
be determined by the dominant methanotrophic phylotypes, but a larger-scale sampling
with activity-based molecular analysis (e.g. RNA or SIP based tools) is required to test
this hypothesis in the future.
The significant $^{13}$C enrichment of soil organic carbon indicated that $^{13}CH_4$-derived
microbial biomass contributed significantly to the turnover of soil carbon. Meta-
analysis indicated microbial biomass represent 0.6-1.1% of total soil organic carbon
(Fierer et al., 2009), but it remained largely unknown about the contribution of methane-
driven microbial food web to soil fertility and quality (Murase and Frenzel, 2007b). Our
results showed no statistically significant difference in net soil $^{13}$C input between low-
pH and high-pH soils during methane oxidation, although type I and type II
methanotrophs employed different strategies for carbon metabolisms (Trotsenko and
Murrell, 2008). For instance, 5-15% of cell biomass carbon in type I methanotrophs
could be derived from $CO_2$ (Trotsenko and Murrell, 2008), and recent study suggests
the proportion up to 60% in type II methanotrophs (Yang et al., 2013). It is noteworthy
that the fresh input of $^{13}CH_4$-derived biomass accounted for up to 1.25% of total organic
carbon in LZ soil (Table S2), implying that the amount of $^{13}CH_4$-C incorporated into
trophic networks comprised a substantial fraction of soil microbial biomass.
Quantitative assessment of soil microbial biomass pools and the relative contribution
of phylogenetically distinct methanotrophs to active carbon pool in soil would be





essential for deciphering the underlying metabolism of methane oxidizers and their
ecological and agricultural importance in paddy fields.

**5 Conclusions**

This study provides evidence for niche differentiation of type I and II
methanotrophs strongly associated with soil pH variation. Low-pH could have likely
selected for type II methanotrophs in paddy soils while type I was favoured in high-pH
soils. The incorporation of $CH_4$-derived carbon into biomass contributed up to 1.25%
of total organic carbon in paddy soil. The fresh input of new carbon from aerobic
methanotrophy played a vital role in the turnover of soil microbial biomass and
subsequent cycling of soil nutrients in support of agricultural sustainability. Nitrogen
fertilization changed methane oxidation rates in five of the soils tested, but the
composition of active methanotrophs was not significantly affected. These results
provide a mechanistic basis for better understandings of community assembly
mechanisms of ecologically important microbial guilds and their possible roles in
agricultural sustainability.
**Data availability.** The raw Illumina sequencing data have been deposited in the
European Nucleotide Archive (ENA) under Ac. No. PRJEB37235 for 16S rRNA genes
and PRJEB40045 for pmoA gene sequences. The sequences of 13C-labelled pmoA
OTUs were deposited to GenBank with accession numbers MK613983-MK613993 and
MK621911-MK621913.
**Author contribution.** JZ and ZJ designed the experiments and JZ carried them out. YC
assisted the bioinformatic analyses. JZ and ZJ prepared the manuscript with
contributions from all co-authors.
**Conflicts of interest.** The authors declare that they have no conflict of interest
**Acknowledgements**. We are grateful to Drs Jing Ma and Dr Yiming Wang and Ms
Lijun Bao at the Institute of Soil Science, CAS, for soil collection. We thank Dr





Baozhan Wang at the Institute of Soil Science for suggestions on the soil chemical characteristics and Mr Zhiying Guo and Wei Gao for technical support in the statistical analysis. We also thank the staff of the Analysis Center at the Institute of Soil Science for technical support, including Ms Rong Huang and Mr Zuohao Ma for Illumina MiSeq sequencing, Ms Deling Sun for the $^{13}$C-atom abundance assay, Ms Yufang Sun for the soil carbon and nitrogen content assay, Mr Ruhai Wang for the ammonia and nitrate-based N content assays, Mr Guoxing Lu for the SOM assay, Mr Hua Gong for the soil metal element measurements and Ms Li Gao for the $SO_4^{2-}$ content assay.

**Financial support.** This work was supported by the National Science Foundation of China (91751204, 41701302 and 41877062), National Key Basic Research Program of China (2015CB150501) and Strategic Priority Research Program of the Chinese Academy of Sciences (CAS) (XDB15040400).





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





Table 1 The sampling locations and chemical characteristics of the rice paddy soils
tested

| Site | Location | pH | SOM (g kg$^{-1}$) | TOC (g kg$^{-1}$) | TN (g kg$^{-1}$) | C/N ratio | Inorganic N (mg kg$^{-1}$) | Cu (mg kg$^{-1}$) | OXC (mmol kg$^{-1}$) |
|---|---|---|---|---|---|---|---|---|---|
| Yu-Xi (YX) | N 24°17' E 102°15' | 5.44± 0.03$^e$ | 27.6± 0.1$^d$ | 15.9± 0.4$^d$ | 1.74± 0.00$^d$ | 9.2± 0.2$^b$ | 85.1± 1.9$^d$ | 1.35± 0.01$^c$ | 18.2± 0.2$^b$ |
| Ying-Tan (YT) | N 28°23' E 116°82' | 5.68± 0.01$^d$ | 19.0± 0.1$^f$ | 13.1± 0.4$^e$ | 1.38± 0.02$^e$ | 9.5± 0.3$^b$ | 204.7± 1.9$^b$ | 1.09± 0.01$^d$ | 11.2± 0.1$^c$ |
| Tao-Yuan (TY) | N 28°55' E 111°27' | 6.10± 0.02$^c$ | 39.5± 0.2$^b$ | 23.1± 0.1$^c$ | 3.15± 0.02$^a$ | 7.3± 0.1$^c$ | 2356.0 ±0.7$^a$ | 1.52± 0.02$^b$ | 6.3± 0.1$^d$ |
| Zi-Yang (ZY) | N 30°05' E 104°34' | 8.02± 0.02$^a$ | 23.2± 0.1$^c$ | 29.9± 0.1$^a$ | 1.93± 0.02$^c$ | 15.5± 0.1$^a$ | 237.8± 0.8$^a$ | 0.52± 0.00$^e$ | 85.7± 2.3$^a$ |
| Chang-Shu (CS) | N 31°33' E 120°42' | 8.02± 0.01$^a$ | 44.7± 0.3$^a$ | 27.7± 0.3$^b$ | 2.90± 0.03$^b$ | 9.6± 0.2$^b$ | 95.5± 1.2$^c$ | 2.07± 0.02$^a$ | 19.3± 0.2$^b$ |
| Lei-Zhou (LZ) | N 20°33' E 110°04' | 7.02± 0.02$^b$ | 19.5± 0.1$^e$ | 13.4± 0.0$^e$ | 1.48± 0.13$^e$ | 9.1± 0.8$^b$ | 86.7± 0.6$^d$ | 0.38± 0.00$^f$ | 19.4± 0.2$^b$ |

Abbreviations: SOM, soil organic matter; TOC, total organic carbon; OXC, soil oxidation capacity
Different letters (a-f) in each row of chemical properties indicate a significant difference between
soils ($P<0.05$).





**Figure 1**

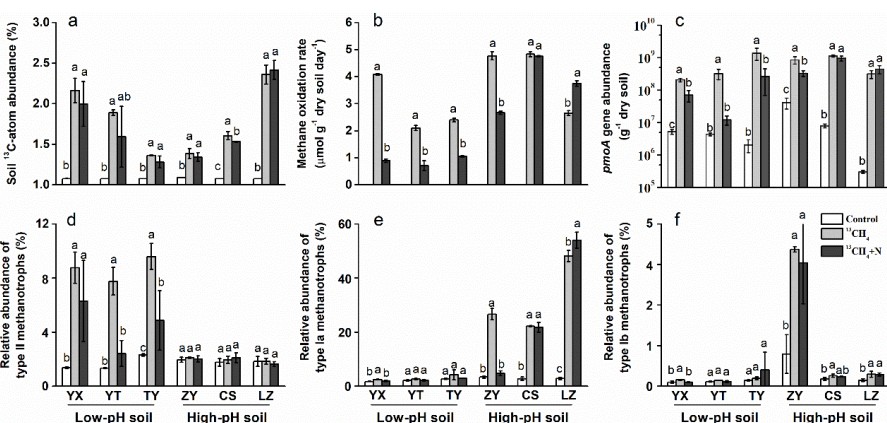


Figure 1 Changes in soil $^{13}$C-atom abundances, methane oxidation rates, community
sizes and compositions of methanotrophs following incubation in soil microcosms. (a)
Soil $^{13}$C-atom abundance was measured to assess methane assimilation in soil
microcosms amended with 5% $^{13}$CH$_4$. (b) The soil methane oxidation rate was
compared between soil microcosms incubated with or without NH$_4$NO$_3$ fertilization. (c)
The *pmoA* gene copy numbers of methanotrophs were estimated using real-time qPCR.
Illumina sequencing targeting 16S rRNA genes was performed at the whole microbial
community level in microcosms, and the relative abundance of type II (d), type Ia (e)
and type Ib (f) methanotrophs was expressed as the ratio of affiliated gene reads to the
total 16S rRNA gene reads in each microcosm. "Control" indicates soil under natural
atmospheric condition. "$^{13}$CH$_4$" and "$^{13}$CH$_4$+N" refer to soil microcosms incubated with
5% v/v $^{13}$CH$_4$ without and with extra NH$_4$NO$_3$ fertilization, respectively. The
designations below the X axis represent the soil sampling sites of rice paddy fields. All
treatments were conducted in triplicate. The error bars represent the standard errors of
the mean of the triplicate microcosms. Different letters above the columns indicate a
significant difference between different treatments in a given soil ($P<0.05$).

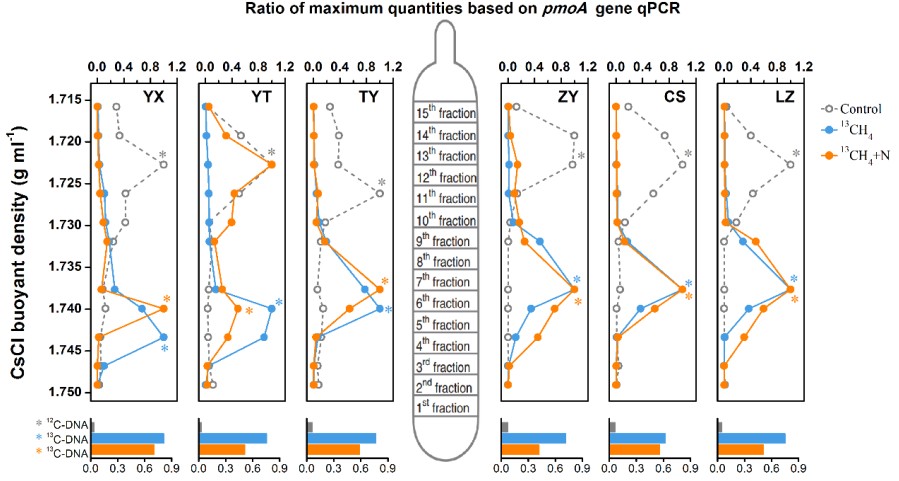

Figure 2 The enrichment of $^{13}$C-labelled methanotrophs based on qPCR of *pmoA* and
sequencing of 16S rRNA genes. The quantitative distribution of *pmoA* genes across
the entire buoyant density gradient of the DNA fractions from soil microcosms
incubated with $^{13}$CH$_4$ compared to the controls. The normalized data are the ratios of
the gene copy number for each DNA gradient to the maximum number for each
treatment. The columns beneath display the relative abundance of methanotroph-
affiliated reads in all 16S rRNA genes in the $^{12}$C-DNA from the control and $^{13}$C-DNA
from $^{13}$CH$_4$-amended soil microcosms, respectively. * represents the DNA fractions
selected for Illumina sequencing.


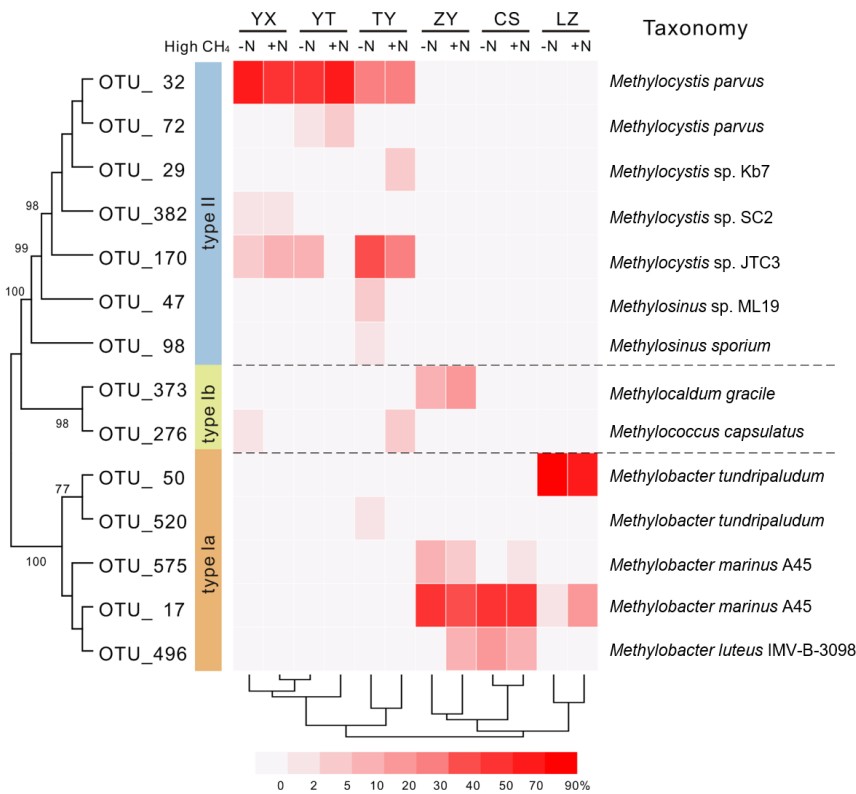

696

Figure 3 Heat map of relative abundances of major $^{13}$C-labelled methanotrophic
OTUs based on *pmoA* gene sequencing. "-N" and "+N" indicate $^{13}$CH$_4$-amended
microcosms without and with nitrogenous fertilization, respectively. Hierarchical
clustering of samples was performed, and phylogenetic relations between different
OTUs are shown by the topology, with bootstrap values >60% indicated at branch
nodes. Each OTU representative sequence is taxonomically annotated to a known
strain from GenBank with the closest phylogenetic relation.





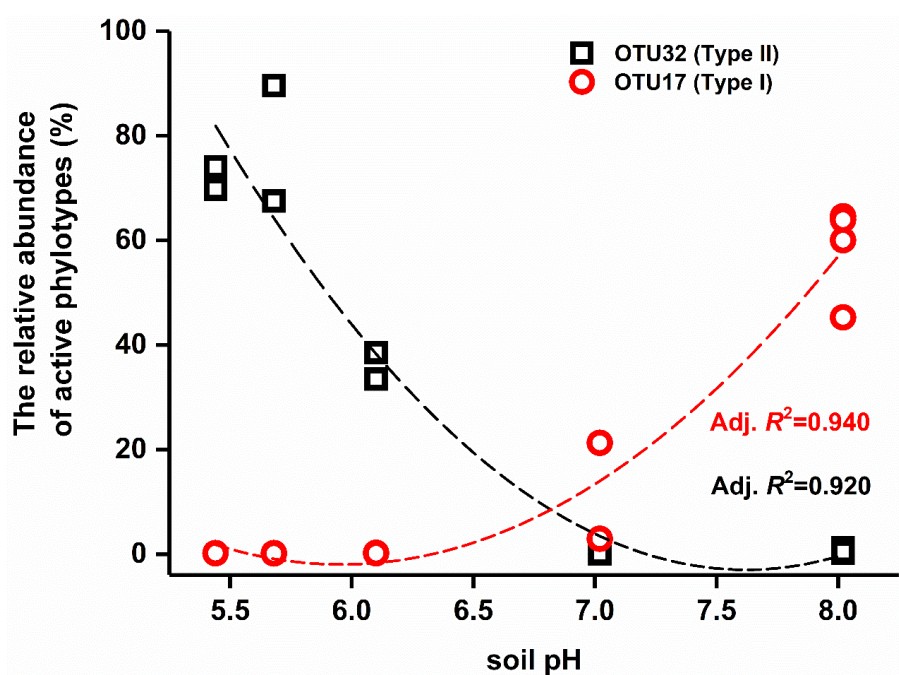

704

Figure 4 The relationship between soil pH and the relative abundance of major [13]C-
labelled methanotrophic phylotypes (OTUs). OTU17 and OTU32 are the same OTUs
displayed in Fig. 3.

708