# Peer review of "3. Author Affiliations"

_Biogeosciences, 2019_

## Referee Comment (RC1) · Anonymous Referee #1 · 28 Dec 2019

This study combined DNA-based stable isotope probing (DNA-SIP) with high-throughput sequencing to identify the taxonomic identities of active methanotrophs in physiochemically contrasting soils from 6 different paddy fields across China. They found that pH is potentially the key driving force selecting for canonical gamma- (type I) and alpha- (type II) methanotrophs in rice paddy soils. It is of interest and innovate to check if the specific functional microbes like methanotrophs are selectively favoured under different pH conditions in natural wetland system. In general, the manuscript is well-written, the results are sound to me, and the discussion are profound. I only have several minor comments as follows: 1. Authors provide solid evidences proving the pH-based ecological coherence of active canonical methanotrophs in paddy soils, but no significant difference of CH4 oxidation was observed between high-pH and low-pH

soils. Why? Please added some discussions. 2. Each microcosm incubation was completed at different time and the longest might be 42 days. Is the microcosm still under oxic condition? Is there any indicators? 3. Subsection 2.8. Sequence data processing and deposition, where is the sequence data deposited? 4. One of the important things for SIP study is to compare the unlabelled and labelled treatments and then identify the labelled microbes. In this study, the control was set as microcosm under natural atmospheric condition. Why not with 12C-CH4 supplementation. Please discuss the possible effect in the discussion.

---

## Author Comment (AC1) · 16 Jan 2020

**Point-by-point response to Reviewer #1's comments:**

This study combined DNA-based stable isotope probing (DNA-SIP) with high-throughput sequencing to identify the taxonomic identities of active methanotrophs in physiochemically contrasting soils from 6 different paddy fields across China. They found that pH is potentially the key driving force selecting for canonical gamma- (typeI) and alpha- (type II) methanotrophs in rice paddy soils. It is of interest and innovate to check if the specific functional microbes like methanotrophs are selectively favoured under different pH conditions in natural wetland system. In general, the manuscript is well-written, the results are sound to me, and the discussion are profound.

**Reply:** Thanks for the positive comments to the present study.

I only have several minor comments as follows:

1. Authors provide solid evidences proving the pH-based ecological coherence of active canonical methanotrophs in paddy soils, but no significant difference of CH4 oxidation was observed between high-pH and low-pH soils. Why? Please added some discussions.

   **Reply**: In this study, despite different community compositions of active methanotrophs between low-pH and high-pH soils, the dominant phylotypes of methanotrophs (type I or II methanotrophs) indeed showed similar oxidation rates. We assume one possible explanation is that the type I and II methanotrophs in the soils might have similar kinetics of methane oxidation, including similar substrate affinity and specific cell activity, according to previous results from pure culture studies (Bedard and Knowles, 1989; Hanson and Hanson, 1996; Calhoun and King, 1998). This interesting point has been added in the manuscript (last paragraph of the Discussion) as following:

   "Despite different compositions of active methanotrophs, there was no fundamental difference of methane oxidation rate between low-pH and high-pH soils, indicating similar methane oxidation rate of type II and I methanotrophs in rice paddy fields. The culture dependent studies have showed similar kinetic traits between these two groups of methanotrophs, including substrate affinity and specific cell activity (Bedard and Knowles, 1989; Hanson and Hanson, 1996; Calhoun and King, 1998), which might explain the similar methane oxidation rates in different soils in the present study.".

2. Each microcosm incubation was completed at different time and the longest might be 42 days. Is the microcosm still under oxic condition? Is there any indicators?

**Reply**: Thanks for the constructive comment. In this study, we completed the incubation at different time points when almost all the amended methane was consumed and then compared the methane oxidation rate based on the average methane consumption quantity per day (Fig. 1b). Indeed, the oxygen concentration is an important factor for methane oxidation and it is possible that lower oxygen concentration might occur with longer incubation period and affect the methanotrophic activity, especially for the microcosms of low-pH soils following fertilization which took more than 30 days to oxidize the majority of the amended methane. Unfortunately, we did not measure the oxygen concentration throughout the incubation, thus to which extent the potential oxygen limitation might affect the methanotrophic rate is not known. However, we have monitored the methane concentrations every 1-3 days throughout the incubation, and the change in methane concentration in each microcosm at during the whole incubation period confirmed the decreased methane oxidation rate following N fertilization in low-pH soils (see the figure below as Fig. S2). Therefore, we believe the present results can provide reasonable argument to the critical point that methane oxidation rate is different in different microcosms and can be affected by nitrogen fertilization.

[Figure]

We have indicated the possible influence of oxygen concentration on the methane oxidation rate in the revised Results section of the manuscript as following: "The lowered methane oxidation rate following fertilization might also suffer from decreased oxygen concentration at the later stage of the microcosm incubation, especially for the fertilized low-pH soil incubations which lasted more than 30 days. However, the temporal changes in the concentrations of headspace methane in the microcosms also showed inhibition by inorganic nitrogen of microbial methane oxidation during incubation of low-pH soils, leading to a prolonged period for consumption of the same amount of methane, particularly for YT soil (Fig. S2)."

3. Subsection 2.8. Sequence data processing and deposition, where is the sequence data deposited?

   **Reply**: Thanks for the comment and we apologize for the mistake here. We did deposit the sequencing in the European Nucleotide Archive (ENA) and the sequence data deposition was described in the section of "Data availability" after "Conclusion" as the journal requires. We have deleted the "and deposition" from the Subsection 2.8 title.

4. One of the important things for SIP study is to compare the unlabelled and labelled treatments and then identify the labelled microbes. In this study, the control was set as microcosm under natural atmospheric condition. Why not with 12C-CH4 supplementation. Please discuss the possible effect in the discussion.

   **Reply**: Thanks for the comment. Indeed, for DNA-SIP study, the active microorganisms are usually revealed by comparing $^{13}$C-labelled and unlabeled microcosms. It is especially important for identifying novel microorganisms with unknown GC content. However, we still think the methane oxidizers revealed from the heavy fractions of $^{13}$C-microcosms reflect mostly the true active methanotrophs in this study despite of absence of $^{12}$C-control, for the following reasons. (1) The goal of this study is to investigate the distribution and activity of well-known canonical methane oxidizers (type I and II) in different pH paddy soils. Based on previous DNA-SIP studies which included both $^{12}$C- and $^{13}$C-methane amended microcosms in our lab and some other research groups (Cai et al., 2016; Daebeler et al., 2014; Dumont et al., 2011; Shiau et al., 2018; Zheng et al., 2014), we expected the $^{13}$C-labelled *pmoA* gene enriched in the SIP fractions with CsCl buoyant density > 1.73 g ml$^{-1}$. In this study, DNA-SIP results from $^{13}$C-methane amended microcosm were consistent to these previous results and the *pmoA* genes accumulated in "heavy" fractions with buoyant densities of approximately 1.735-1.745 g ml$^{-1}$. (2) Although $^{12}$C-control was not used in this study, we performed the DNA-SIP fractionation on the microcosms under natural atmospheric condition, and the *pmoA* genes accumulated in the "light" fractions with a buoyant density of 1.717-1.726 g ml$^{-1}$, which is also similar to the results of $^{12}$C-control microcosms from the previous studies mentioned above. These *pmoA* genes should represent the background methanotrophs in the soils without being $^{13}$C-labelled. (3) Furthermore, the sequencing of *pmoA* genes from "heavy" fractions of $^{13}$C-methane amended microcosms showed that the methane oxidizers were dominantly canonical gamma- (*Methylobacter* and *Methylocaldum*) and alpha- (*Methylocystis*) methanotrophs, which do not have particularly high G+C content and should not be abundantly detected in the target "heavy" fractions (with density of 1.735-1.745 g ml$^{-1}$) without successful incorporation of $^{13}$C into their DNA. (4) However, we cannot rule out that there were a small amount of unlabeled *pmoA* genes "drifting" to the "heavy" fractions during the process of

DNA fractionation, but it is confident to conclude that at least the dominant phylotypes revealed in the "heavy" fractions in this study were the truly labelled, active methanotrophs.

We added the information of similar buoyant density compared to previous studies in the revised Results section and discussed the lack of $^{12}$C-methane treatment and possible effect in the Discussion section as follows:

In Results 3.4, we compared the CsCl density range in our study with previous studies by adding "The *pmoA* genes in the $^{13}$CH$_4$-amended microcosms accumulated in the heavy DNA fractions with a CsCl buoyant density of approximately 1.735-1.745 g ml$^{-1}$, which was within the same range as in previous studies (Shiau et al., 2018;Cai et al., 2016), while the *pmoA* gene abundance in the control treatments peaked only in the light DNA fractions with a buoyant density of 1.717-1.726 g ml$^{-1}$."

In Discussion paragraph 2 we added "Although we cannot rule out the possibility that there were some unlabeled *pmoA* genes "drifting" to the heavy fractions during the process of DNA fractionation, since the absence of microcosms with $^{12}$C-methane amendment for background calibration prevents precise calculation of relative abundance of $^{13}$C-labeled microbes, it is confident to conclude that the dominant phylotypes revealed in the heavy fractions in this study represented the truly labeled and the most active methanotrophs."

**Reference:**

Bedard, C., and Knowles, R., 1989. Physiology, biochemistry, and specific inhibitors of CH4, NH$_4^+$, and CO oxidation by methanotrophs and nitrifiers. Microbiological Reviews, 53, 68-84.

Cai, Y., Zheng, Y., Bodelier, P.L.E., Conrad, R., Jia, Z., 2016. Conventional methanotrophs are responsible for atmospheric methane oxidation in paddy soils. Nature Communications 7, 1–10.

Calhoun, A., and King, G. M., 1998. Characterization of root-associated methanotrophs from three freshwater macrophytes: Pontederia cordata, Sparganium eurycarpum, and Sagittaria latifolia. Applied and Environmental Microbiology, 64, 1099-1105.

Daebeler, A., Bodelier, P.L.E., Yan, Z., Hefting, M.M., Jia, Z., Laanbroek, H.J., 2014. Interactions between Thaumarchaea, Nitrospira and methanotrophs modulate autotrophic nitrification in volcanic grassland soil. ISME Journal 8, 2397–2410.

Dumont, M.G., Pommerenke, B., Casper, P., Conrad, R., 2011. DNA-, rRNA- and mRNA-based stable isotope probing of aerobic methanotrophs in lake sediment. Environmental Microbiology 13, 1153–1167.

Hanson, R., and Hanson, T., 1996. Methanotrophic bacteria Microbiological Reviews, 60, 439-471.

Shiau, Y.J., Cai, Y., Jia, Z., Chen, C.L., Chiu, C.Y., 2018. Phylogenetically distinct methanotrophs modulate methane oxidation in rice paddies across Taiwan. Soil Biology and

Biochemistry 124, 59–69.

Zheng, Y., Huang, R., Wang, B.Z., Bodelier, P.L.E., Jia, Z.J., 2014. Competitive interactions between methane- and ammonia-oxidizing bacteria modulate carbon and nitrogen cycling in paddy soil. Biogeosciences 11, 3353–3368.

---

## Referee Comment (RC2) · Anonymous Referee #2 · 19 Jan 2020

Zhao et al. investigated the active aerobic methanotrophs in six different paddy soils. They do find the key roles of pH in mediating the niche differentiation of aerobic methanotrophs. Methylocystis-affiliated type II methanotrophs were most active in soils with pH (5.44-6.10), while Methylobacter/Methylosarcina-affiliated type I methanotrophs were most active in soils with pH (7.02-8.02). In addition, no significant changes in 13C-labelled methanotrophic community compositions was detected with the fertilization of ammonium nitrate. The study is interesting and the manuscript is well written. The questions are appropriately introduced and explored and the results are well presented. Therefore, I would recommend the manuscript for publication. However, it should be noted these results were obtained from rice paddy soils. The consistency in other soils should be further studied. For example, deng et al. 2016

(Ref: Identification of active aerobic methanotrophs in plateau wetlands using DNA stable isotope probing) found that both Methylocystis and Methylobacter methanotrophs were dominated in methane consumption in Dangxiong peatland soils with a pH of 6.1. Therefore, the authors should consider more comprehensively when interpreting the results. In line 217, why did the authors cluster the pmoA genes at a nucleotide level of 93/ similarity. In fact, we usually define 93/ similarity at amino acid level and corresponding to methanotrophic species (Reference: Lüke C. and Frenzel P., 2011 Potential of pmoA Amplicon Pyrosequencing for Methanotroph Diversity Studies). If the authors want to cluster the pmoA sequences at a nucleotide level, Wen et al., updated the pmoA gene cutoff at nucleotide level to 86/ corresponding to 97/ similarity of 16S rRNA gene (Reference: Wen et al., 2016 Evaluation and update of cutoff values for methanotrophic pmoA gene sequences).

———————————————————

---

## Author Comment (AC2) · 24 Jan 2020

**Point-by-point response to Reviewer #2's comments:**

Zhao et al. investigated the active aerobic methanotrophs in six different paddy soils. They do find the key roles of pH in mediating the niche differentiation of aerobic methanotrophs. Methylocystis-affiliated type II methanotrophs were most active in soils with pH (5.44-6.10), while Methylobacter/Methylosarcina-affiliated type Imethanotrophs were most active in soils with pH (7.02-8.02). In addition, no significant changes in 13C-labelled methanotrophic community compositions was detected with the fertilization of ammonium nitrate. The study is interesting and the manuscriptis well written. The questions are appropriately introduced and explored and the results are well presented. Therefore, I would recommend the manuscript for publication.

**Reply:** We thank the reviewer for general positive comments and recommendation for publication of this study.

However, it should be noted these results were obtained from rice paddy soils. The consistency in other soils should be further studied. For example, deng et al. 2016 (Ref: Identification of active aerobic methanotrophs in plateau wetlands using DNA stable isotope probing) found that both Methylocystis and Methylobacter methanotrophs were dominated in methane consumption in Dangxiong peatland soils with a pH of 6.1. Therefore, the authors should consider more comprehensively when interpreting the results.

**Reply:** Thanks for the advice. The present study was designed to reveal the potential niche specialization of methane oxidizers influenced by soil pH in the rice paddy fields, which represent one of the most typical anthropogenic wetland ecosystems. We fully acknowledge that the results from this study might not reflect the situation of methanotrophic assembly in soils and sediments from other ecosystems including natural peatlands, in which other factor(s) might be equally or more critical. To prevent overstatement of the role of pH in methanotrophic activity and distribution, in the revised manuscript, (i) we will present our conclusions more carefully and restricted our results, conclusions and discussions "in rice paddy soils". (ii) We will also include the reviewer's advice in the revised discussion as following: "However, the consistence of our conclusion of the pH-based ecological coherence of methanotrophs in other environments should be further studied, since previous studies also revealed co-dominance of type I and II methanotrophs in low-pH natural peatlands (Deng et al., 2016;Esson et al., 2016)"

In line 217, why did the authors cluster the pmoA genes at a nucleotide level of 93/ similarity. In fact, we usually define 93/ similarity at amino acid level and corresponding to methanotrophic species (Reference: Lüke C. and Frenzel P., 2011 Potential of pmoA Amplicon Pyrosequencing for Methanotroph Diversity Studies). If

the authors want to cluster the pmoA sequences at a nucleotide level, Wen et al., updated the pmoA gene cutoff at nucleotide level to 86/ corresponding to 97/ similarity of 16S rRNA gene (Reference: Wen et al., 2016 Evaluation and update of cutoff values for methanotrophic pmoA gene sequences).

**Reply:** Thanks for this informative comment. Our basis for choosing 93% similarity is from a previous literature describing use of similarity cutoff at the range of 87-95% for *pmoA* genes (van de Kamp et al., 2019). After carefully reading the literatures from the reviewer' comment, we must admit that clustering *pmoA* OTUs at 87% or 86% similarity is theoretically well-founded and was more commonly used. However, in this study, we chose higher similarity (93%) to cluster *pmoA* genes considering that higher resolution clustering of functional markers is useful to detect spatial patterning in microbial assembly as suggested by van de Kamp et al. (2019). Clustering at 93% similarity would result in more observed methanotrophic genotypes (OTUs) than at 86-87% cutoffs, thus allowing detection of potentially fine community composition difference between different soils. Additionally, this similarity cutoff should not change our main conclusion here that either type I or type II methanotrophs dominating active methane oxidation. We will add the following information in the revised manuscript to support our methodology: "Clustering *pmoA* genes at 93% similarity rather than commonly used 86-87% similarity (Wen et al., 2016;Degelmann et al., 2010) is useful to detect spatial patterning in microbial assemblage as suggested previously (van de Kamp et al., 2019)". We fully respect the reviewer's suggestion and concern here, and hope our explanation is adequate to justify our results.

**Reference:**

Degelmann, D. M., Borken, W., Drake, H. L., and Kolb, S.: Different atmospheric methane-oxidizing communities in European beech and Norway spruce soils, Appl Environ Microbiol, 76, 3228-3235, 10.1128/AEM.02730-09, 2010.

Deng, Y., Cui, X., and Dumont, M. G.: Identification of active aerobic methanotrophs in plateau wetlands using DNA stable isotope probing, FEMS Microbiol Lett, 363, 10.1093/femsle/fnw168, 2016.

Esson, K. C., Lin, X., Kumaresan, D., Chanton, J. P., Murrell, J. C., and Kostka, J. E.: Alpha- and Gammaproteobacterial Methanotrophs Codominate the Active Methane-Oxidizing Communities in an Acidic Boreal Peat Bog, Appl Environ Microbiol, 82, 2363-2371, 10.1128/AEM.03640-15, 2016.

van de Kamp, J., Hook, S. E., Williams, A., Tanner, J. E., and Bodrossy, L.: Baseline characterization of aerobic hydrocarbon degrading microbial communities in deep-sea sediments of the Great Australian Bight, Australia, Environ Microbiol, 21, 1782-1797, 10.1111/1462-2920.14559, 2019.

Wen, X., Yang, S., and Liebner, S.: Evaluation and update of cutoff values for methanotrophic pmoA gene sequences, Arch Microbiol, 198, 629-636, 10.1007/s00203-016-1222-8, 2016.